# Measures to Promote Olive Grove Biomass in Spain and Andalusia: An Opportunity for Economic Recovery against COVID-19

**Jesús Marquina, María José Colinet and María del P. Pablo-Romero ***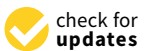

Department of Economic Analysis and Political Economy, Faculty of Economics and Business Sciences, University of Seville, Ramon y Cajal 1, 41018 Seville, Spain; jmarquinadelaossa@gmail.com (J.M.); mjcolinet@gmail.com (M.J.C.)
* Correspondence: mpablorom@us.es; Tel.: +34-954-557-611; Fax: +34-954-557-629

**Abstract:** Olive grove biomass presents an opportunity to reduce greenhouse gases and meet the sustainability objectives set by Europe. Given the relevance of this organic matter, this paper analyzes the evolution and current situation of the regulations that regulate olive grove biomass as a source of energy in Europe, in Spain and in Andalusia. Likewise, its effect on the evolution of the use of biomass in the Andalusian region, especially the olive grove, is analyzed. The analysis is novel, since there are no previous studies that reveal this type of information for the olive grove biomass sector. The results show that, as of 2005, the development of biomass for thermal and electrical uses is favorable, reaching the objectives set by the PASENER 2007–2013. However, this situation is reversed as of 2012, with the abolition of the feed-in tariff system for renewables. Besides this, the olive grove biomass sector faces other obstacles such as the cost of residue collection and the few incentives for this sector. The reorientation of the measures, in order to enhance this energy source, would generate a positive effect for the economy of the region that has been affected by COVID-19.

**Keywords:** renewable energy; biomass; olive grove; energy regulations; energy plans; Spain; Andalusia

---

## 1. Introduction

The current health crisis caused by COVID-19 has led to a bleak outlook throughout the global economy [1]. The world economy has been hit hard, experiencing a contraction of 3.5% in 2020 [2]. This indicates a recession much worse than that of the 2008 crisis (GDP contraction of 1.3%) [3]. Faced with this current economic framework, much emphasis has been placed on a green recovery as a means to stimulate the economy [4]. In fact, the UN has established the promotion of economic recovery plans that take into account the climate and nature crisis. Likewise, the International Renewable Energy Agency (IRENA) urges governments to face the challenge of economic recovery by taking advantage of the progress made in renewable energies, so as not to lose sight of the fight against climate change and the commitment to sustainability [5]. In this way, it is intended that renewable energies play a key role in economic recovery, guaranteeing sustainability and energy security, creating employment and strengthening resilience to protect people's health and well-being.

Faced with this international panorama, olive grove biomass, as a renewable energy source, plays a key role in this green recovery that is being implemented. This organic matter presents an opportunity for Mediterranean countries, which is where the highest concentration of this crop exists and, therefore, where the greatest generation of residues originate.

Thus, the olive grove is considered to have high economic and energy value. Currently, it has a presence in countries where it was traditionally considered unimaginable that

---

these could exist, such as China, Australia, Latvia and Finland. However, despite its greater expansion to various countries around the world, 80% of the surface of olive groves is concentrated in the Mediterranean basin [6]. Likewise, olive oil industries also have a greater presence in Mediterranean countries, reaching 94% of the world's olive oil production [7]. The great wealth generated by this crop and the agro-processing sector in this geographical area, means that, every year, large quantities of biomass are generated that can be used for energy use. This reveals the importance this crop has for these countries, since its waste presents an alternative clean energy source. At the same time, it contributes to improvement in climate change (one of the main objectives worldwide).

Olive grove biomass as an energy source is regulated by the different rules governing this sector. In Europe, its regulation began from 1997, with the approval of the Treaty of Amsterdam (where the principle of sustainable development was included in the Community Objectives) and the preparation of the White Paper on Energy [8]. With the latter, the main aim was to promote the use of renewable energies. The first measures proposed by the White Paper on Energy focused on improving competitiveness, security of supply and protection of the environment. Its fiscal and financial measures included a favorable tax treatment for renewable energies, subsidies for the start-up of new production plants and financial incentives for consumers [9]. From that moment, companies related to the agricultural and olive sector (mainly olive mills, table olive industry, olive pomace extractors and a small number of growers) began to have greater incentives to develop tasks related to use of waste, with the purpose of producing energy, mainly for self-consumption.

In parallel, in the same year, the approval of the Agenda 2000 invited the promotion of renewable energy sources. It was also proposed that biomass should be developed by all available means in the agricultural, fiscal or industrial fields, which encouraged the Member States to grant aid for their support [10]. This background influenced the development and progress of biomass as a source of renewable energy. Its relevance, at European level, is evidenced by its high level of production (102,615 ktoe, which represents 44.67% of the total production of renewable energies in 2019) [11]. Among the main European producers are Germany, France, Italy, Sweden, Finland and Poland, with 57.56% of the total production in 2019 [12]. The main consumers of biomass are the Nordic and Baltic countries, together with Austria, and headed by Finland [13].

Within this group, no Mediterranean basin olive grove cultivation countries are predominant. In this sense, at the European level, Spain did not occupy a privileged place in the production of primary energy from biomass. However, it does stand out for its ability to obtain this resource. In 2020, the Spanish territory had 50.6 million hectares cultivated (5.43% corresponded to olive groves) and had a solid and numerous agro-processing sector producing large amounts of biomass each year [14,15]. In spite of this, in 2018, the biomass potential was 17,287 ktoe (18.36% of the European total) and consumption was 5444 ktoe (1314 ktoe for electrical uses and 4130 ktoe for thermal uses) [16,17].

Within Spain, in 2018, Andalusia represented 17.1% of the potential biomass (2963 ktoe) which is equivalent to 16.2% of the region's primary energy consumption [16]. The region had 8.76 million agricultural hectares in 2020, of which 1.66 million were occupied by olive groves [14]. In addition, the olive crop has continued to expand (8.1% since 2004) [14]. Thirteen of the seventeen Andalusian power generation plants traditionally use the remains of the oil production industry (olive pomace, extracted olive pomace, olive stones and olive leaves) as fuel for the olive industry itself [18].

The relevance of the olive sector in the Andalusian region, its prospects for growth in the immediate future, and its capacity to generate renewable energy from its waste, justify the interest of this study. Thus, the objective of this paper is to analyze the effects that the main approved regulations have had on the development of olive grove biomass in Andalusia, and whether their effects have been positive or negative. In the same way, this paper aims to encourage policies aimed at promoting olive grove biomass as an energy source for thermal and electrical uses. In addition, it is also intended that the use of this energy source represents a possible alternative to stimulate the economy, which has



been greatly aggravated by the health crisis caused by COVID-19. Therefore, the launch of a "Green Recovery" is proposed. For this, the plans and programs, laws, and other regulations approved for this purpose and its main consequences are analyzed in a triple scenario: European, Spanish and regional. With this aim, the following procedure is followed for the development of this paper. In the first place, the main regulations that regulate biomass production for thermal and electrical energy purposes in Europe, in Spain and in Andalusia are compiled and systematized, with special mention of olive grove biomass. Thus, the main measures and aims of these regulations are detailed. Second, the effect of these regulations on the evolution and level of development of olive grove biomass in Andalusia is analyzed. It should be said that similar analyses have been carried out previously, but referring to other renewable energy sources or referring to other countries. Proof of this are the studies by Pablo-Romero et al. (2013) on measures to promote solar thermal energy in Spain [19] or that of Bouznit et al. (2020) on measures to promote renewable energy for electricity generation in Algeria [20]. However, this paper presents a novel study, since as far as we know, there are no previous studies that analyze the regulations that have been regulating the use of olive grove biomass in recent years and its effects on the development of its capacity, particularly in Andalusia.

It is important to point out that in the European and Spanish areas, there are numerous regulations that may also have an indirect influence on the development of olive grove biomass for electricity production. In this paper we refer only to those that have a direct impact.

The results of the analysis allow us to determine in what way these measures have influenced the development of biomass from olive groves in Spain and Andalusia, as a renewable energy source for electrical and thermal uses. The analysis is relevant, since it may allow the orientation of the policies to be improved in the main olive-growing country and region, with extension to other Mediterranean countries, also affected by COVID-19.

This paper highlights different unsolved problems that this energy sector has in Spain and, especially, in Andalusia. In this sense, the main problems are related to: current regulations, high cost of labor and transport and the limited incentives for this sector to correct the aforementioned obstacles. Thus, this paper offers an opportunity to open the debate on a problem of great magnitude that paralyzes the development of olive grove biomass in a region where its potential is high.

The main contributions presented in this paper can be used to boost olive grove biomass in those places where this crop is grown. Thus, this paper provides a set of information not previously collected in any previous paper. In this way, the main novelties that this paper contributes are:

- Compilation of the main regulations regarding biomass, making special references to aspects related to olive grove biomass.
- Representation and critical assessment of the main energy data of olive grove biomass at European, national and regional level.
- Discussion on the main contributions of this renewable energy source to the regional economy.

The structure of this paper is as follows. Following this introduction, Section 2 describes the methodology used. The results are given in Section 3. Section 4 discusses the results and, finally, Section 5 details the latest conclusions.

## 2. Methodology

Figure 1 shows the methodological procedure of this paper:

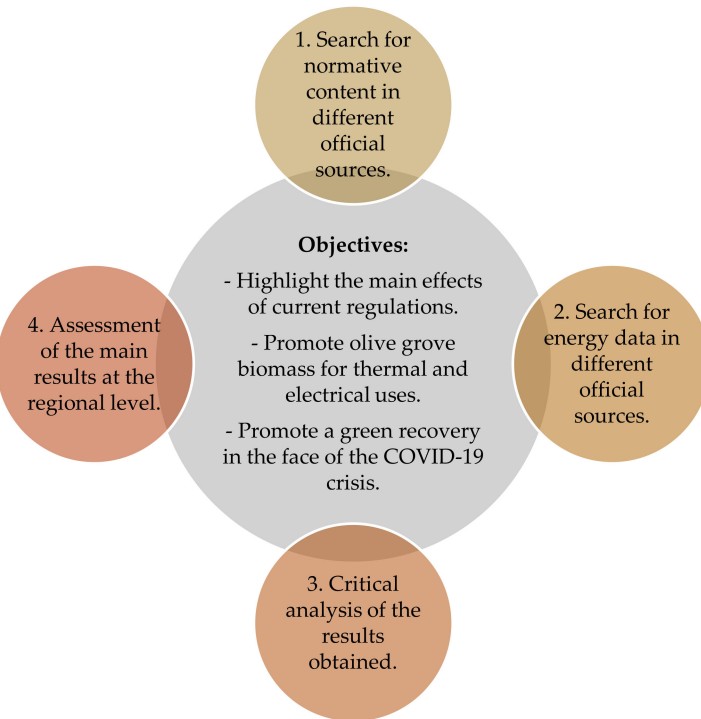

**Figure 1.** Outline of the research methodology process.

First, the different regulations (energy plans, laws and incentive programs) are compiled and analyzed. From these, the most relevant aspects that affect the biomass and, especially, the olive grove biomass, stand out. The analysis is carried out from the first approved renewable energy policies (1997) until 2020. The main sources of information search are: the website of the European Commission, BOE (Official State Gazette), BOJA (Official Gazette of the Junta de Andalusia) and Ministries of Spain and Andalusia.

Second, energy data referring to biomass and olive grove biomass are collected. Likewise, these data are collected for the period 1997–2020. The data are obtained from different official sources such as Eurostat, the European Environment Agency, IDAE (Institute for Diversification and Energy Saving) and the Andalusian Energy Agency.

Third, a critical analysis is made with the results obtained at the European, national and regional level.

Fourth, the main results are evaluated at the regional level.

Finally, it should be noted that the reason that justifies the search means used to collect information is the official nature and quality of the different publications in certain high-impact journals.

## 3. Results

The results of this paper can be divided into three parts. In each of them, the main regulations and energy plans, approved in Europe, in Spain and in Andalusia, are analyzed. This highlights what effects the policies have had on the development of olive grove biomass, and what opportunities for increased waste usage would mean for the economic recovery after COVID-19.

### 3.1. Olive Grove Biomass Regulations in Europe

The cultivation of the olive tree occupies more than 11 million hectares throughout the world. Almost 7 million (60%) are concentrated in Europe, more specifically in the countries of the Mediterranean basin [6]. As a result, the EU is a world leader in olive oil production, with an average production of 2.1 million tons per year (68% of world production) [21]. Most of this production is concentrated in the Mediterranean area, dominated by Spain, Italy and Greece (70% of European production), Spain being the main producer of olive oil

(with a production of 1.12 million tons during the 2019–2020 season) [22,23]. In this regard, the importance of the olive sector justifies the relevance of this crop in its contribution to the production of renewable energies. Therefore, the waste generated by this crop has been the subject of a great deal of interest in the development of different European regulations related to renewable energies.

### 3.1.1. Communications from the European Union

The first measures to boost olive biomass in Europe took place with the approval of the White Paper on Energy in 1997 [COM (1997) 0599]. Since then, as shown in Table 1, the EU has published a range of documents related to its position in favor of reducing climate change, and its commitment to the use of renewable energies. In each of these documents, biomass is the subject of special attention, as this energy source is the most used within the European territory [11].

**Table 1.** Communications from the EU regarding the promotion of the use of renewable energies.

| Name | Year |
| --- | --- |
| White Paper "Energy for the future: renewable sources of energy" [COM (1997) 0599] | 1997 |
| White Paper on environmental liability [COM (2000) 66 end] | 2000 |
| "Biomass Action Plan" [COM (2005) 628 end] | 2005 |
| Green Paper "A European Strategy for Sustainable, Competitive and Secure Energy" [COM (2006) 105 end] | 2006 |
| "Forest Action Plan" [COM (2006) 302] | 2006 |
| "Renewable energy road map—Renewable energies in the 21st century: building a more sustainable future" (Action Plan 2007–2009). | 2007 |
| Progress Report on the implementation of the Biomass Action Plan [COM (2009) 0192] | 2009 |
| Report on sustainability requirements for the use of solid and gaseous biomass sources in electricity, heating and cooling [COM (2010) 0011 end] | 2010 |
| "Clean Energy for All Europeans" [COM (2016) 860 end] | 2016 |
| The European Green Deal [COM (2019) 640 end] | 2019 |
| Annex on guidelines for trans-European energy infrastructure [COM (2020) 824 end] | 2020 |

Source: Own elaboration from EUR-Lex.

In 2005, the European Commission launched, for the first time, a "Biomass Action Plan" [COM (2005) 628 final]. This plan established a series of measures aimed at increasing the development of this type of energy from wood, waste and agricultural crops [24]. It also aimed to reduce dependence on fossil fuels, reduce greenhouse gas emissions and stimulate economic activity in rural areas. All this was due to the fact that biomass represented more than half (from 44% to 65%) of the renewable energy consumed in Europe. In this regard, the European Commission identified three sectors in which the biomass resource should be a priority: heat production, electricity production and transport. In addition, all the points of this plan were related to a set of measures, carried out by the Common Agricultural Policy (CAP), where much of the aid was aimed at promoting energy crops and the energy use of agricultural by-products and waste. In the case of the olive grove, the established aid contemplated certain agro-environmental practices, among them, the active use of pruning waste.

One year later, the European Commission presented the "Forest Action Plan" [COM (2006) 302] as part of the development of the Forestry Strategy for the European Union (1998), which also includes biomass from the olive groves [25]. This Plan contemplates four main objectives to optimize sustainable management and the multifunctional role of European Union forests:

- Increase long-term competitiveness.
- Improve and protect the environment.
- Contribute to a better quality of life.
- Encourage communication and coordination with the aim of increasing coherence and cooperation at different levels.

Thus, it was intended to promote the use of forest biomass for energy production, focusing on the development of markets for wood pellets and chips.

Later, a Progress Report on the implementation of the Biomass Action Plan was carried out in 2009 [COM (2009) 0192]. It recognized the deviation from compliance with the 2010 objective, regarding renewable energy sources. This deviation was due to the increase in energy consumption within the EU and the insufficient development of renewables [26]. Among the conclusions, it highlighted the need to act prudently in the use of biomass as a source of renewable energy, and proposed an adequate supervision system.

In 2010, the European Commission issued the "Report . . . on sustainability requirements for the use of solid and gaseous biomass sources in electricity, heating and cooling" [COM (2010) 0011 final]. It recommended that Member States that had introduced, or would introduce, national sustainability systems in relation to solid and gaseous biomass, ensure that these systems were equal to those established in the Renewable Energy Directive. This was intended to encourage the production and sustainable use of biomass, a well-functioning internal biomass market and the elimination of obstacles to the development of bioenergy [27].

In the same way, it is important to underline the Commission's Release on "Clean Energy for All Europeans" [COM (2016) 860 final], in 2016. This established that only the efficient conversion of biomass into energy would receive public support, those that presented favorable characteristics and met adequate sustainability criteria. This was intended to avoid the possible adverse effects that uncontrolled use of biomass could cause for the climate [28]. In this sense, olive cultivation had favorable characteristics that placed it in a better position, with respect to other crops. This was due to the fact that the olive is an organized arboreal crop, whose biomass is controlled by the growth needs of the plants [29]. In short, although it had to comply with the EU's sustainability criteria, it was part of a more favorable position than biomass from natural forest areas [28].

Finally, it is worth highlighting the European Green Pact [COM (2019) 640 final], which includes a series of measures related to the climate and the environment. Thus, the priority objective is to promote that the EU is the first climate-neutral continent in the world by 2050 [30]. Likewise, the Annex on guidelines for trans-European energy infrastructure [COM (2020) 824 final] aims to expand the European energy infrastructure, in order to address the fragmentation of interconnections between Member States in their isolation from the networks of gas and electricity, secure and diversify the Union's energy supplies, sources and supply routes, as well as increase the integration of renewable energy sources [31]. Both regulations, recently approved, directly promote the use of olive grove biomass for energy purposes.

### 3.1.2. European Directives

Directive 2001/77/EC [32] was the first to promote compliance with the objectives in the EU. Later, Directive 2003/54/EC was approved and established definitions applicable to the electricity sector in general [33]. Although there was still no express mention of the biomass sector, there was general regulation of renewables. This encouraged the EU's energy consumption to grow steadily for years. Between 2000 and 2007, it was approximately 25 TWh per year. Likewise, biomass boiler installations increased significantly from 2004, making applications for heating and SHW supplied through pellets, a common practice in many European countries [13].

In 2009, with the entry into force of Directive 2009/28/EC, the objective to reach 20% of final energy consumption from renewable sources by 2020, was established. From that moment, biomass began to be mentioned in the regulatory field, with the establishment of sustainability criteria for biofuels and bioliquids [34]. This Directive was revised in June 2018, when sustainability criteria for the use of biomass was addressed, and in the case of residual biomass it stated the following [35]:

- Biomass should not come from land of high value in terms of biodiversity, nor from peatlands that are not drained.

- Forest biomass should not be produced unsustainably.
- The reduction of emissions must be at least 80% for the production of electricity, heating and cooling from biomass in installations, following its start-up on January 1, 2021, and 85% as of January 1, 2026.
- Power generation plants over 20 MW must always use cogeneration technology.

Finally, in December 2018, Regulation (EU) 2018/1999 came into force. It established a series of measures in line with The Paris Agreement 2015, and the specific objectives of the Union for 2030, on energy and climate. In this way, the objectives for renewable energy and energy efficiency are set for the year 2030 at 32% and 32.5%, respectively. These objectives are also included in Directive (EU) 2018/2001 and Directive (EU) 2018/2002, respectively [36]. Both objectives will be revised in 2023 and can only be updated to raise them and not to reduce them. Furthermore, this new regulation prevents charges being applied to self-consumption. Regarding transportation, it has been established that at least 14% of the fuel used must come from renewable sources by 2030, and first-generation biofuels with a high risk of "indirect change in land use", should not count towards the renewable use objectives from then on. In conclusion, these can place the olive grove biomass sector in a more favorable position, in such a way that it can be translated into a future increase in its uses.

### 3.1.3. The Olive Grove in the Evolution of Biomass in Europe

The promotion of renewable energies in Europe has caused the proportion of their use in total energy consumption to increase in recent years. In the 2005–2015 decade, their proportion in EU consumption almost doubled, from 9% to approximately 17% [37]. However, despite the greater efforts by the EU, fossil fuels are still the dominant energy source in Europe, representing 72.6% of the energy mix.

Biomass stands out for its production levels, which makes it the leading renewable energy in the EU. Figure 2 shows the evolution of biomass production in Europe, over the total of renewables produced. Since 1997, it has had the highest percentage of use, followed by hydroelectric (11.97% of the total) and wind (13.74% of the total). Biomass currently provides approximately 4% of the total EU energy supply. However, and despite the fact that in 2019 it represented 44.67% of renewable production, participation has been decreasing slightly over time [11]. The production of biomass in the EU is mainly used for residential heat (83%) and, to a lesser extent, for combined generation (CHP) of heat and electricity (17%) [16].

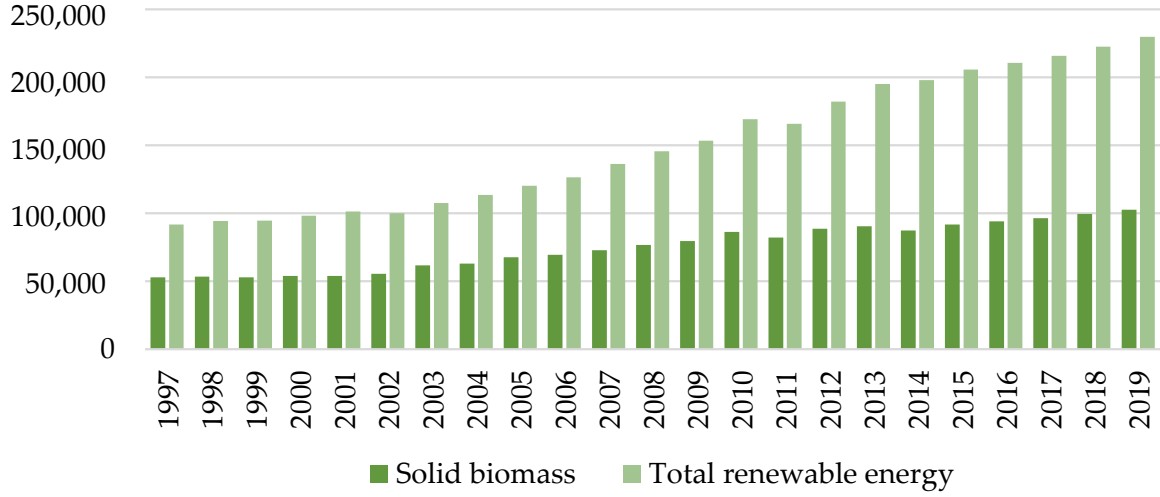

**Figure 2.** Evolution of the production of solid biomass and the set of renewable energies in Europe during the period 1997–2019 (units: in ktoe). Source: Own elaboration from Eurostat.

The olive grove has favorably contributed to this situation. The oil mills generated 9.6 million tons of biomass per year (mainly, olive pomace, olive leaves and olive stones). For its part, olive grove pruning generated 11.8 million tons of biomass, of which only 30% were used [38]. In this way, in Europe, this crop reached an approximate volume of 13.1 million tons of biomass per year. The olive grove biomass is most often used for electricity generation (47%) and for thermal generation (33%). Regarding the use of thermal energy, the main consumer is the olive oil industry (oil mills) and extraction, followed by the agro-processing sector that does not relate to olive oil, and the domestic sector [38]. Olive cultivation also plays a fundamental role in the advancement of this type of energy, which is due to the fact that its main by-products represent an important way to generate electrical and thermal energy, especially in Mediterranean countries [39].

### 3.2. Measures for the Promotion of Olive Grove Biomass in Spain

Spain has considerable agricultural wealth, which, among other advantages, provides an important opportunity to generate renewable energy. This country ranks second among the countries with the greatest agricultural area utilized in the EU (with 13.3%, and behind France which has 15.9%) [40], where the olive groves occupy 2.75 million hectares [14]. Most of this crop (60%) is concentrated in the Guadalquivir valley (Andalusia), mainly in the provinces of Jaen (22%) and Cordoba (13%) [41].

### 3.2.1. Renewable Energy Plans

In Spain, the production and use of olive grove biomass have been influenced by the different energy plans related to renewable energies that have the purpose of promoting their production and use and, consequently, meeting the objectives set by Europe. The main objectives of each of these plans are shown in Table 2. This shows that it was not until 1999, when a specific renewable energy plan was approved, because the 1991 plan, although it included the planning of all energy sources, only introduced a specific chapter dedicated to the promotion of renewable energies.

Since 1999, the three plans approved have adhered to the guidelines set by Europe, with increasing demands for renewable energy. The purposes and objectives set for biomass are contained in each of these plans, where the olive plays a fundamental role.

The National Energy Plan—PEN 1991–2000 basically intended to paralyze the nuclear programs, approved in previous Plans, and to promote the expansion of natural gas and renewable energies [42]. On the other hand, the Plan for the Promotion of Renewable Energy—PFER 2000–2010 began to forge an energy structure that was increasingly beneficial for renewables. From this moment on, the first targets for biomass were highlighted .

**Table 2.** Main renewable energy plans in Spain.

| Approval Date. | Energy Plans | General Objectives | Specific Objectives for Biomass |
|---|---|---|---|
| 1991 | National Energy Plan (PEN) 1991–2000 | - Minimization of costs. <br> - Energy diversification. <br> - Self-supply. <br> - Environmental Protection. <br> In this way, it plans to subsidize the actions aimed at promoting and improving energy efficiency, committing to renewable energy. | The use of biomass for energy use is encouraged, replacing fossil fuels. |
| 1999 | Plan for the Promotion of Renewable Energy (PFER) 2000–2010 | - Natural gas becomes the source with the greatest growth, up to 22.50% of the total in 2010. <br> - The electricity generation structure consists of: 33% natural gas, 28.4% renewable energy, 19.4% nuclear, 15% coal and 4.1% petroleum products. | It is planned to increase its production for 2010 by 6 million ktoe. Of the total, 0.9 would correspond to thermal uses in the final consumption sectors and 5.1 would be destined to the generation of electricity. In this way, the contribution established, up to the year 2000, is multiplied by 30. For this, it is intended to encourage different economic sectors and households. |
| 2005 | Renewable Energy Plan (PER) 2005–2010 | - To cover at least 12% of the total energy consumption in 2010 with renewable sources. <br> - To achieve a minimum of electricity generation with renewable (29.4%) on the gross national consumption of electricity. <br> - To achieve a minimum of biofuels (5.75%) in relation to gasoline and diesel consumption in transport, in accordance with Directive 2003/30/EC. | The targets for biomass in 2010 are set at 1695 MW of installed power for electrical uses. In this way, production would be 11,822.6 GWh plus a production of 582,514 toe/year in thermal biomass. |
| 2011 | Renewable Energy Plan (PER) 2011–2020 | - To achieve a 20% share of energy from renewable sources in gross final consumption by 2020. <br> - To achieve a 10% share of energy from renewable sources in energy consumption in the transport sector by 2020. | For the solid biomass, an objective of 1187 MW is established for 2020, and an increase of 383 MW of power is expected for thermal uses in 10 years. |

Source: Own elaboration from BOE, IDAE [42–45].

Subsequently, during the validity of the Renewable Energy Plan—PER 2005–2010, an ambitious target for biomass was established. This was set at 1695 MW of installed power for electrical uses, aimed at generating 11,822.6 GWh/year. However, only 32% of the target for solid biomass was reached. In this way, the development of solid biomass for electric power generation was small, despite the important advantages that the main by-products of olive groves could contribute to the energy market. In 2010, the primary biomass energy over the total represented 4.8%, mainly due to its use for thermal applications. In general, the renewables percentage, as primary energy over the total, was 11.3%, so the target of 12% set in the PER 2005–2010 was almost reached [46]. The renewable energies experiencing the greatest growth were wind, solar photovoltaic and solar thermoelectric.

Later, with the Renewable Energy Plan—PER 2011–2020, the requirements of the objectives continued to be increased. Its purpose was to adapt to the new European objectives, established in Directive 2009/28/EC. In the case of biomass, it continued to show a deficit in its development, reaching installed power targets below those highlighted in the previous Plan. Thus, the incipient attempts to encourage its use for energy use and the granting of subsidies (BIOMCASA, GIT, PAREER-CRECE) to promote its projects, had not been enough of an incentive for the development of the sector.

In parallel, and in order to fulfill the commitments with Europe in the field of renewable energies, different Action Plans have been launched from Spain. These were

framed in the obligations established by Directive 2009/28/EC. The National Renewable Energy Action Plan—PANER 2011–2020 Plan is currently in force, reflecting the objectives established by Europe [47]. For the biomass sector, a target of 1587 MW was set, of which 1187 MW would correspond to solid biomass, and 400 MW to biogas. In Spain, there are only 677 MW of biomass installed for electrical uses, which demonstrates the scarce impact of current policies.

Furthermore, The Spanish Bioeconomy Strategy—2030 Horizon, which serves as support for the fulfilment of the objectives of the current Energy Plan, is also relevant. It aims to move towards a society that is less dependent on non-renewable resources, to slow down the process of climate change. The foundation on which it is based is the science–economy–society triangle. In this way, it is a guarantee that all economic agents will collaborate, with the purpose of reducing the polluting effects and generating economic value [48]. Following the same path, the Report of the Committee of Experts of the Parliament (finalized in April 2018) established the basis for an efficient, sustainable and low carbon energy transition. To do this, the environmental and economic impacts were considered when designing the energy policy. The report included several alternatives that analyzed the combination of different sources of energy (nuclear, hydraulic, thermal, coal, combined cycles and renewable sources). In addition, it evaluated the objective of renewable penetration, according to different levels of interconnection with the European continent, and the contribution of mobility and energy efficiency policies [49].

### 3.2.2. Regulations on the Use of Olive Grove Biomass for Energy in Spain

In Spain, renewable energies are driven by current electrical regulations at all times. In these regulations, there is no specific section for olive sector biomass. However, it does fall within different headings, such as that of biomass from agricultural or industrial waste.

In general, the regulatory framework for the electricity sector has been subject to various changes. Thus, as shown in Table 3, from 1994 to present, three Laws relating to the electricity sector have been approved. Law 24/2013 is currently in force, which means a substantial change in the treatment of renewable energies. With this Law, among other issues, renewables cease to have priority in the discharge of energy, compared to other technologies. As of this moment, they only enjoy an equal economic offer in the daily electricity market. In addition, the structure of the system that had been established for renewable energy sources and cogeneration (included in the special regime for installations of up to 50 MW), and the rest of fossil and nuclear technologies (ordinary regime), has been broken. As a result, all energy sources (renewable and non-renewable) compete under equal conditions, abandoning the differentiation between the powers of the ordinary regime and the special regime. Thus, the system of previously established premiums (determined by Ministerial Order IET/1045/2014) was eliminated.

The most developed renewable energies have been wind, solar photovoltaic and solar thermoelectric. During 2007–2013, these experienced strong growth in MW capacity. In 2018, they amounted to 30,473 MW installed, which represented 60% of the national total [17].

With regard to biomass, it should be noted that the implementation of this energy source for electrical uses has been insufficient, when compared with other renewable sources. The biomass capacity for such uses is only 677 MW of capacity. This represents 1.34% of the total renewable power in Spain. However, biomass for thermal uses is in a better position, given its greater uses in the energy market (especially in the household sector), it has 8297 MW of installed capacity [17].

**Table 3.** Main Laws of the electricity sector.

| Law | Period of Validity | Description |
|---|---|---|
| Law 40/1994, On Management of the National Electricity System. | 1994–1998 | The concept of a special regime is consolidated, and the security of electricity supply is guaranteed at the lowest possible cost and adequate quality. |
| Law 54/1997, On the Electricity sector. | 1998–2013 | Distinguishes production in the ordinary regime from the special regime. The economic remuneration framework is identified for each of these models of electricity generation, including biomass. The system of premiums for electric power generated by special regime installations is the system chosen for promotion of these facilities. |
| Law 24/2013, On the Electricity sector. | 2013 to date | The basic purpose is to establish regulation of the electricity sector. Guaranteeing supply with the necessary levels of quality and at the lowest possible cost, ensuring economic and financial sustainability of the system and allowing an effective level of competition within the electricity sector. All this within the principles of environmental protection of a modern society. The premium system is replaced by a competitive system in auctions (with some exceptions). The profitability of these facilities refers to the ten-year State obligations. |

Source: Own elaboration from BOE [50–52].

In parallel to the Laws listed in Table 3, several Royal Decrees have been developed during the period 1994–2017 (Table 4.). These have had some impact on the development of biomass at the national level. In general, from 1998 to 2013, an incentive system for "feed-in tariff" electricity production was established. Subsequently, in 2016, auctions for renewable facilities were established, based on offers on the cost of investment and operation of the plants.

**Table 4.** Main Royal Decrees on renewable energies for electricity generation.

| Period of Validity | Royal Decree | Description | Aspects Relevant to Biomass |
|---|---|---|---|
| 1994–1998 | Royal Decree 2366/1994 On production of electrical energy by hydraulic, cogeneration and other facilities, supplied by renewable energy sources. | Regulating the electrical energy of the "*Special Regime*", where biomass is the accepted renewable energy source. | There is no specific mention of the biomass energy sector. |
| 1998–2004 | Royal Decree 2818/1998 On electric energy production facilities supplied by resources or sources of renewable energy, waste and cogeneration. | A premium is determined for facilities below 50 MW using renewable non-consumable and non-renewable energy, biomass, biofuels or agricultural, livestock or service waste, as primary energy. | A distinction is made between primary biomass (in the case of olive groves it corresponds to biomass from olive tree waste), and secondary biomass (corresponding to waste from the olive oil manufacturing industry). This distinction is reflected in the compensation established for each type, with a 5.07 pesetas/kWh (0.03 EUR/kWh) premium for the first, and a 4.70 pesetas/kWh (0.028 EUR/kWh) premium for the second. |
| 2004–2007 | Royal Decree 436/2004 On the methodology for updating and systematization of the legal and economic regime for the production activity of electric power under the special regime is established. | A system, based on the freedom of choice of the owner of the installation, is defined: - Sale of the production or surplus electric power to the distributor. A fee in the form of a regulated tariff is received, unique to all programming periods, based on the production market price. - Sale of the production or surplus directly on the daily or futures markets or through a bilateral contract. The negotiated market price is received, plus a participation incentive and a premium, if the actual installation has the right to receive it. Regardless of the compensation mechanism chosen, holders of special regime facilities are guaranteed a reasonable remuneration for their investments. | Article 37 establishes tariffs, premiums and incentives for biomass facilities. The fee for participating would be 80%, the premium 30% and the incentive 10%, of the average or reference electricity tariff in each year. |
| 2007–2013 | Royal Decree 661/2007 Regulating the production activity of electric power in special regime. | Regulates the activity of energy production in special regime and repeals Royal Decree 436/2004. The basic scheme is maintained. The double option of rewards is preserved. Determines a premium complementing the remuneration regime of biomass and/or biogas co-combustion plants in thermal power plants of the ordinary regime. | An installed capacity of 1317 MW is established as an objective for installations using biomass as fuel. The equivalent powers of biomass or biogas in co-combustion installations are not considered within the objectives of reference installed power. |

**Table 4.** *Cont.*

| Period of Validity | Royal Decree | Description | Aspects Relevant to Biomass |
|---|---|---|---|
| 2010 to date | Royal Decree-Law 14/2010 On the establishment of urgent measures to correct the electricity sector tariff deficit. | Alleviates the asymmetric effects caused by the economic crisis. Measures are: obligatory tolls on renewable energies (0.5 EUR/MWh), application of percentage economic participation for producers under ordinary and special regimes to mitigate cost overruns and, for photovoltaic installations, limits equivalent operational hours with rights to preferential economic regime. | No special measures are specified or established for the biomass sector. |
| 2012 to date | Royal Decree-Law 1/2012 By which pre-allocation procedures for compensation are removed and economic incentives for new installations for the production of electricity from cogeneration, renewable energy sources and waste are eliminated. | Economic incentives granted to new facilities producing renewable energy are suppressed and new start-up facilities are paralyzed. | Article 3.3. establishes that the government can establish statutorily specific economic regimes for certain installations under the special regime (which includes biomass). The right to the perception of a specific economic regime is permitted and, where appropriate, certain obligations and rights regulated under Sections 1 and 2 of Article 30 of Law 54/1997. |
| 2013 to date | Royal Decree-Law 2/2013 On urgent measures in the electrical system and the financial sector. | The objective is to correct economic imbalances occurring in the electrical system. The reduction of special regime costs is highlighted. Consumers are asked to contribute to economic recovery through consumption and investment. | The reference premium value for all groups (including biomass) is modified, having a value of 0 c EUR/kWh. |
| 2013 to date | Royal Decree-Law 9/2013 By which urgent measures are adopted to guarantee financial stability of the electrical system. | Permits Government approval of a new legal and economic regime for existing electric power production facilities, from renewable energy sources, cogeneration and waste. Thus, it is about giving back to those companies in the renewable sector that are considered efficient, based on the facilities having reasonable profitability set by regulations. | Includes measures, at the general level of the renewable energy sector, without including specific measures on biomass. |
| 2014 to date | Royal Decree 413/2014 On the regulation of production of electrical energy from renewable energy sources, cogeneration and waste. | New remuneration regime for renewables is established. Based on the perception of income obtained from the sale of electricity to the market, plus an additional remuneration (calculated using a series of standardized parameters according to existing technologies in the market and included in Ministerial Order 1045/2014). | Installations that use biomass should send information on the ratio of the types of fuels used (through a certification system), indicating the annual amount used, in tons, per year and the average PCI, in kcal/kg, of each of them. |

**Table 4.** *Cont.*

| Period of Validity | Royal Decree | Description | Aspects Relevant to Biomass |
|---|---|---|---|
| 2015 to date | Royal Decree 900/2015 On the regulation of the administrative, technical and economic conditions of the modalities of electricity supply with self-consumption and production with self-consumption. | First regulation of electricity self-consumption in Spain. | No specific reference to biomass. |
| 2015 to date | Royal Decree 947/2015 By which a call is established for the granting of the specific remuneration regime to new installations for production of electricity from biomass at the peninsula's electrical system and wind technology installations. | First auction launched for the allocation of specific remuneration regime to electricity production facilities from wind and biomass technology. | The power for new biomass installations located on the Iberian Peninsula is set at 200 MW. |
| 2017 to date | Royal Decree 359/2017 By which a call is established for the granting of the specific remuneration regime to new installations for the production of electrical energy from renewable energy sources in the peninsular electrical system. | Auctions for 2017, treat all technologies neutrally, without differentiation. | Beneficial for the most developed energy sources (wind and photovoltaic) and represents a further setback in the advance of biomass. |

Source: Own elaboration from Noticias Jurídicas, BOE [53–64].

With Royal Decree 2366/1994, the renewable energy sector began to be regularized, making a general mention of biomass. However, it was not until 1998, with Royal Decree 2818/1998, when this sector began to be consolidated.

Subsequently, Royal Decree 436/2004 aimed to continue down the path already started by the previous Royal Decree, but with an added advantage: the application of a system based on the free will of the seller, which consisted of choosing between two options when selling its production on the energy market. Likewise, it was intended that, by the year 2010, one-third of the demand for electricity would be covered by high energy efficiency technologies and renewable energies (without increasing the cost of electricity production) [55]. Later, with the entry into force of Royal Decree 661/2007, the renewable energy sector continued to be reinforced, in this case, with an increase in economic incentives for electricity generation (feed-in tariff). The effects that occurred in the biomass sector during the term of this Royal Decree (2007–2013) were positive. This was due to the installed capacity of biomass for electrical uses being increased by 74.73%, with a total installed biomass power of 657 MW.

After the economic crisis of 2008, the national panorama began to change. With the instability of the electricity sector, several approved regulations paralyzed development in the renewable energy sector. The economic crisis had a strong negative influence on the *tariff deficit*. This began in 2000 and, in 2005, the deficit growth was accentuated, reaching more than EUR 30,000 million in 2011 [65]. Thus, with the approval of Royal Decree-Law 14/2010, these were required to contribute economically to the special regime energies in the electricity system to prevent cost overruns. However, this was intensified with the approval of Royal Decree-Law 1/2012, which introduced a serious cut in existing premiums. At the same time, this paralyzed incentives for new facilities. As of that moment, the concept of special regime and ordinary regime was held in check.

Subsequently, in 2013, Royal Decree-Law 2/2013 and Royal Decree-Law 9/2013 were approved in order to correct the problems of the electricity and financial sectors. Equally, this meant a barrier for the development of renewable energies and, in particular, for olive grove biomass. All this led the renewable energy sector to lose advantages over fossil fuels, which remain the most competitive in the national energy market. In addition, it adversely affected the fulfilment of the objectives, set by Europe in Directive 2009/28/EC, and included in the 2011–2020 Renewable Energy Plan.

Currently, Royal Decree 413/2014 regulates the production of electricity from renewable energy sources. Its approval supposes a radical change for the renewable energies that have begun to lose some privileges. This new remuneration system was based on the application of an additional remuneration (calculated through a series of standardized parameters, established by Ministerial Order 1045/2014) after the sale of electricity on the market.

The new remuneration system, based on the auction system, has not been a boost to the development of biomass projects. So far, there has been only one specific biomass auction, regulated by Royal Decree 947/2015. This fixed the installed power of 200 MW for new biomass installations on the Iberian Peninsula. As a result, electric power using biomass, in the period 2013–2018, had only increased by 3.04%, standing at 677 MW. Regarding the other technologies, the following auctions were assigned: wind 4608 MW, photovoltaic 3910 MW and others 19 MW.

In parallel to electrical regulations, different regulations have been approved that affected the development of biomass as an energy source for thermal uses. It is worth mentioning that Royal Decree 314/2006, which implemented the Technical Building Code [66], replaced the old Regulation of Thermal Installations in Buildings (RITE) and gave a boost to the renewable sector, especially biomass for thermal uses. The approval of this Royal Decree implied a commitment to energy efficiency and environmental balance. In addition, it encouraged the incorporation of renewable energy facilities in building processes and gave greater prominence to the European requirements, highlighted by Directive 2002/91/EC. Five basic requirements were developed, aimed at achieving a rational use of the energy

necessary for the use of buildings, and ensuring that part of this consumption came from renewable sources. Likewise, with the approval in that same year of Royal Decree-Law 7/2006, urgent measures were adopted in the energy sector, and the transaction costs of competition (CTC's) were repealed. In this way, the variation of the special regime premiums were disconnected from the average electrical, or reference, tariff [67].

### 3.2.3. The Olive Grove in the Biomass Evolution in Spain

Figure 3 shows the evolution of total installed renewable power, as well as the power in biomass, for both thermal and electrical uses, from the period 1998–2004 to the period 2013–2020. The periods mark the time that each of the main Royal Decrees, related to the production of renewable energies, had been in force (RD 2818/1998, RD 436/2004, RD 661/2007 and RD 413/2014). It can be observed that the installed biomass power, like the other renewable energies, had greater growth during the validity of PER 2005–2010, with increases of more than 70% with respect to the period 2004–2007, when the PFER 2000–2010 was in force. In addition, it should be noted that although the data in Figure 3 only refer to the periods of validity of each of the Royal Decrees, in all years the power of biomass has been increasing, reaching the values of 2020 (677 MW).

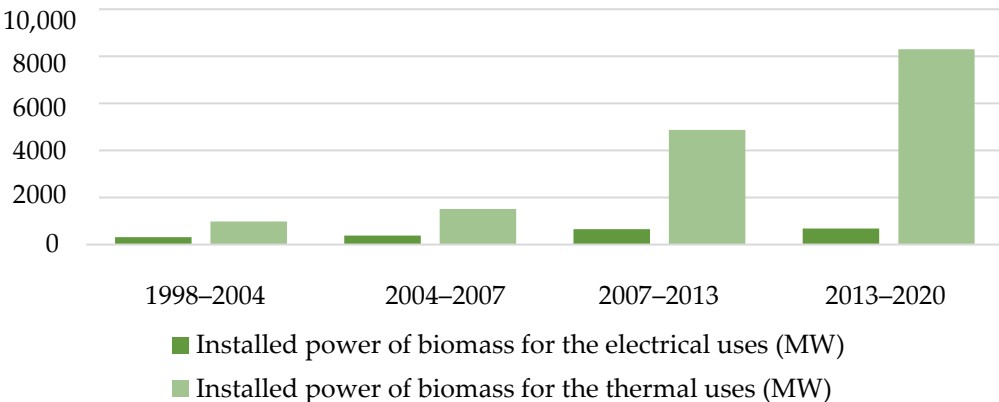

**Figure 3.** Evolution of the electrical and thermal power of biomass installed in Spain during the period 1998–2020 (units: in MW). Source: Own elaboration from IDAE.

However, and despite this growth of biomass, the effects of the measures to promote it for electrical use could be considered insufficient, at least when compared with other technologies. Currently, the installed power of renewable energies in Spain stands at 50,356 MW, with the system generating 104,607 GWh, representing 38.1% of total gross electricity production. Wind, with 22,990 MW (45.66% of the total installed capacity), and hydraulics, with 18,963 MW (37.66% of the total installed power), are those that generate the most energy in Spain, with 47% and 35%, respectively. Meanwhile, biomass, with 677 MW of installed capacity (1.34% of the total nationally) only generated 5%.

The high potential of biomass in Spain is due, in part, to the cultivation of the olive and the waste generated by the agro-food olive oil industries. As can be seen in Figure 4, the territory has 2.75 million hectares occupied by this crop. These are mainly concentrated in Andalusia (60.43%), although they are also present in Castile-La Mancha (16.12%) and Extremadura (10.45%) [14]. Thus, Spain has large areas of olive cultivation where around 282.70 million olive groves are concentrated. This generates large amounts of biomass per year, as a result of the different olive grove pruning processes (mainly, thin branches and firewood) [68]. Firewood produced from pruning is often used for domestic heating in rural areas. The thinner pruned wood is used less frequently due to high collection costs, resulting in it being burned in the open field.

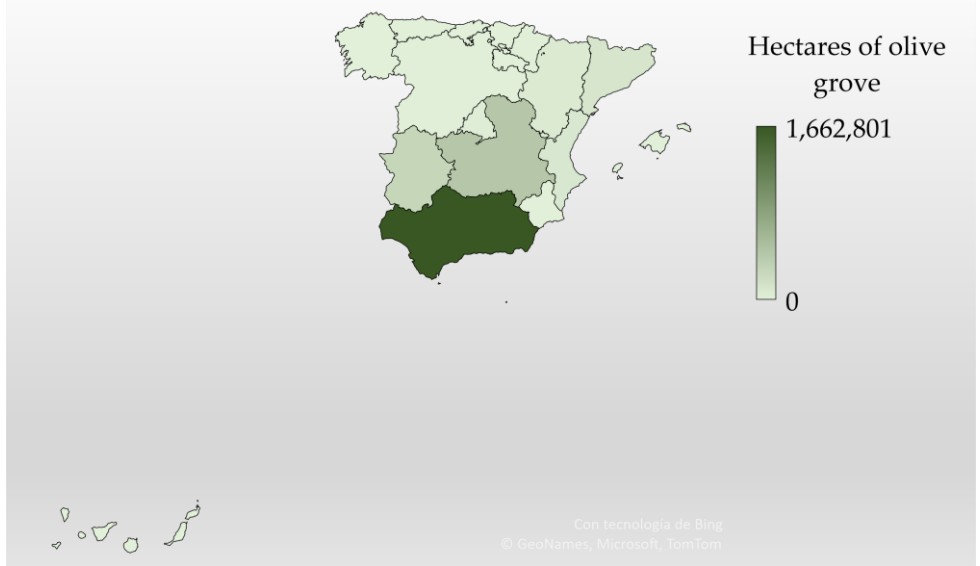

**Figure 4.** Distribution of the cultivated area of olive groves in Spain (units: number of hectares). Source: Own elaboration from ESYRCE.

Furthermore, Spain has a large number of agro-food industries, of which 1756 correspond to olive mills (where olive oil is obtained), 487 to the table olive industry (consumable olives) and 63 to olive pomace extractors (extraction of residual oil by chemical processes, producing olive-pomace oil) [23]. These industries are responsible for producing the various products derived from olives, such as table olives, olive oil or olive-pomace oil. These production processes produce different by-products: olive pomace, extracted olive pomace, olive stones and the remains of leaves and branches which are used to generate electrical and thermal energy, often used for self-consumption by the industry [18].

The importance of the olive sector in biomass production in Spain can be demonstrated in different mathematical applications offered and published by various studies. Thus, the outline of the article "Report on the availability of Biomass Sources in Spain: vineyards and olive groves" shows that, in Spain, olive grove pruning generates approximately 7 million tons per year [69]. The largest contribution is made by Andalusia (57%), followed by Castile La-Mancha (17%), Extremadura (11%) and Catalonia (5%). These four regions produce 90% of the total biomass generated by olive grove pruning in Spain [70]. For its part, the olive oil industry generates 4.75 million metric tons of olive pomace (50% is destined to the manufacture of pomace olive oil), 1.45 million metric tons of olive stones and 0.78 million metric tons of extracted olive pomace. The table olive industry produces 0.06 million metric tons of olive stones per year [69].

Despite its high potential, the lower growth rate of biomass, compared to other technologies, may be due to the lower incentives received by the sector (when compared to those for other renewable sources such as wind, solar photovoltaic and solar thermoelectric), and the greater complexity of the treatment this resource requires.

Table 5 shows the results of compensation payments (feed-in tariffs) during the 2014–2020 period. It can be observed that the most prioritized technologies are those of cogeneration, solar photovoltaic, solar thermoelectric and wind, which are precisely those that have advanced most during this period. Biomass, for its part, follows these energy sources, but with a major difference. In relation to the remuneration of generated electricity it can be observed that biomass has higher remuneration than wind, consistent with the increased degree of technological maturity of the latter technology. However, the rewards are much higher in the case of photovoltaic technology, in spite of major technological developments in wind technology and its worldwide expansion. This, together with insufficient incentives to the agricultural sector for the use of waste, has contributed to the

reduced development of olive biomass. Basically, the remuneration does not cover the costs of collection and transport and does not offer an acceptable return for the investor [29].

**Table 5.** Result of the annual compensation payments for production installations of renewable energies, cogeneration and waste during the 2014–2020 period.

| Technology | Year | Feed-in Tariff (M EUR) | Feed-in Tariff (EUR/MWh) |
|---|---|---|---|
| Cogeneration | 2020 | 917.86 | 39.80 |
| | 2019 | 1312.89 | 49.72 |
| | 2018 | 1233.65 | 46.89 |
| | 2017 | 1147.40 | 44.80 |
| | 2016 | 871.04 | 36.82 |
| | 2015 | 1173.20 | 50.53 |
| | 2014 | 1702.00 | 64.30 |
| Solar PV | 2020 | 2354.15 | 182.73 |
| | 2019 | 2407.83 | 282.64 |
| | 2018 | 2441.97 | 326.51 |
| | 2017 | 2493.40 | 323.52 |
| | 2016 | 2439.30 | 321.26 |
| | 2015 | 2441.60 | 315.37 |
| | 2014 | 2822.00 | 374.67 |
| Solar TE | 2020 | 1244.46 | 279.03 |
| | 2019 | 1285.73 | 263.96 |
| | 2018 | 1297.12 | 297.23 |
| | 2017 | 1313.40 | 260.60 |
| | 2016 | 1263.00 | 257.86 |
| | 2015 | 1262.50 | 261.12 |
| | 2014 | 1441.00 | 244.82 |
| Wind power | 2020 | 1208.63 | 23.45 |
| | 2019 | 1413.89 | 26.71 |
| | 2018 | 1475.33 | 30.47 |
| | 2017 | 1450.80 | 30.74 |
| | 2016 | 1256.90 | 26.60 |
| | 2015 | 1253.10 | 26.87 |
| | 2014 | 1823.00 | 36.68 |
| Hydraulics | 2020 | 64.93 | 14.73 |
| | 2019 | 76.77 | 18.95 |
| | 2018 | 94.91 | 15.16 |
| | 2017 | 81.60 | 28.52 |
| | 2016 | 75.10 | 16.43 |
| | 2015 | 67.70 | 16.08 |
| | 2014 | 164.00 | 36.55 |
| Biomass | 2020 | 330.52 | 76.33 |
| | 2019 | 311.98 | 89.09 |
| | 2018 | 315.16 | 90.98 |
| | 2017 | 306.80 | 88.49 |
| | 2016 | 274.30 | 82.87 |
| | 2015 | 258.30 | 76.15 |
| | 2014 | 283.00 | 66.11 |
| Waste | 2020 | 86.30 | 32.58 |
| | 2019 | 109.12 | 36.05 |
| | 2018 | 118.34 | 36.71 |
| | 2017 | 119.40 | 35.34 |
| | 2016 | 104.90 | 31.29 |
| | 2015 | 102.50 | 28.95 |
| | 2014 | 81.00 | 27.75 |

**Table 5.** *Cont.*

| Technology | Year | Feed-in Tariff (M EUR) | Feed-in Tariff (EUR/MWh) |
|---|---|---|---|
| Waste treatment | 2020 | 250.46 | 68.39 |
| | 2019 | 223.59 | 74.36 |
| | 2018 | 171.98 | 66.61 |
| | 2017 | 152.80 | 63.19 |
| | 2016 | 86.10 | 52.73 |
| | 2015 | 120.50 | 78.66 |
| | 2014 | 416.00 | 97.40 |
| Other technologies | 2020 | 1.17 | 68.40 |
| | 2019 | 1.10 | 57.89 |
| | 2018 | 0.19 | - |
| | 2017 | 0.19 | - |
| | 2016 | 0.16 | 1454.55 |
| | 2015 | 0.56 | 140.00 |
| | 2014 | - | - |

Source: Own elaboration from National Markets and Competition Commission (CNMC).

*3.3. Measures for the Promotion of Olive Grove Biomass in Andalusia (Spain)*

The cultivated area of olive groves in Andalusia occupies 1.66 million hectares (34.04% of the agricultural surface), which is equivalent to 60% of the Spanish olive oil producing area and 30% of that of the EU [14,29]. Likewise, the industrial sector that surrounds this crop, at a national level, is predominantly concentrated in this region where 48% of the oil mills (844), 71% of the olive pomace extractors (45) and 45% of the table olive industry are located (219) [41]. In these industries, products such as olive oil, olive-pomace oil (representing 76% of national production and 42% of the Community's production) and table olives (representing 67% of the national production and 45.9% of the Community's production) are sold. The favorable characteristics of this sector have, over the years, forged a solid market, both nationally and internationally [29]. In turn, the olive grove presents other advantages to the Andalusian region, since the different by-products that are obtained from it, and from its agro-processing sector, have a high renewable energy value. This forms an alternative energy to fossil fuels. This advantage offers the possibility of reducing greenhouse gas emissions to the atmosphere and contributes to the improvement of climate change. In addition, in 1999, the primacy of its by-products boosted the start-up of the first biomass power generation plant in Spain, in Palenciana (Cordoba, Andalusia). Currently, this power generation plant (El Tejar Autogeneración) uses extracted olive pomace (a by-product obtained from the olive-pomace oil manufacturing process) as fuel to generate energy [18].

3.3.1. Effects of the Different Energy Plans for Olive Grove Biomass in Andalusia

In Andalusian regional planning, olive grove biomass has received special attention, as this energy source has developed more in this territory than in Spain's other Autonomous Communities. This is due to the abundant presence of this crop and the high energy richness of its by-products.

Since 1995, Andalusia has launched four plans in the field of energy planning: Energy Plan of Andalusia—PLEAN 1995–2000; Energy Plan of Andalusia—PLEAN 2003–2006; Andalusian Plan for Energy Sustainability—PASENER 2007–2013; and the Andalusian Energy Strategy 2020. The four plans shared the commitment to improve energy efficiency and increase the use of renewable energy, over and above national and community objectives. Specifically, the PLEAN 2003–2006, was intended to cover at least 15% of the total demand for primary energy in the region by 2010, three percentage points more than those highlighted by Europe for that year (12%).

In each of the energy plans, biomass and olive growing in particular played an important role. Table 6 shows the most relevant aspects.

**Table 6.** Plans to promote the use of renewable energies in Andalusia.

| Approval | Plans | General Objectives | Relevant Aspects to Biomass |
|---|---|---|---|
| Prepared in 1994 by the Society for Energy Development of Andalusia, for the Ministry of Economy and Finance. | Energy Plan of Andalusia (PLEAN) 1995–2000 | Its objective was that, by 2000, renewable energies would contribute 7.6% of the total primary energy. | It aimed to increase biomass by 202 ktoe/ear and develop cogeneration and electricity and gas infrastructure in the region. |
| Decree 86/2003, of April 1 | Energy Plan of Andalusia (PLEAN) 2003–2006 | Its purpose was to ensure that, by 2010, 15% of energy demand corresponded to renewable energies, a significant proportion of this objective being obtained in 2006. | In the case of biomass for electrical uses, the objective was to achieve an installed capacity of 250 MW (the figure of 164 MW was reached in 2006) by 2010. Furthermore, due to expected high growth of biomass for thermal uses, the *Programme to Promote the Use of Biomass—Probiomass* was launched to encourage facilities. |
| Decree 279/2007, of November 13 | Andalusian Plan for Energy Sustainability (PASENER) 2007–2013 | Its objective was, in 2013, that 18.3% of primary energy consumption would correspond to renewable energy and 32.2% to gross production of electricity with renewable energy. | With regard to biomass, it was intended to achieve 256.0 MW and 649.0 ktoe, for electrical and thermal uses, respectively. This represented 21.13% of the total renewable energy. |
| Agreement of October 27, 2015, of the Government Council | Energy Strategy of Andalusia 2020 | Five objectives were proposed for 2020: to reduce the trend of primary energy consumption by 25%; to provide 25% of the final gross energy consumption from renewable energies; to decarbonize energy consumption by 30% with respect to 2007 values, to consume 5% of the electricity generated from renewable sources; and improve the quality of the energy supply by 15%. | Two Action Plans were agreed to carry out this Strategy. They included specific measures for the development of a biomass value chain, its energy use and the development of bio-refineries. |

Source: Own elaboration from Consejería de Innovación, Ciencia y Empresa [71–74].

All these plans have incentivized change in the region's energy model. In this way, the consumption of clean energies has been increasing. Currently, energies such as solar, wind and biomass account for 38.8% of installed electrical power [75]. This is equivalent to 40.4% of the electricity consumed by Andalusians.

Together with these plans, other strategies have also been addressed in parallel that have served as supports for the fulfilment of these objectives. Among them are the "Autonomic strategy in the face of climate change" approved in 2002 and the "Andalusian Strategy for Sustainable Development: Agenda 21 of Andalusia" approved in 2004. These were created with the purpose of reducing greenhouse gas emissions. They promoted renewable energies, the reduction of energy dependence, energy saving, the establishment of instruments for energy improvement and the construction of energy efficient homes.

Similarly, in 2007 the "Andalusian Climate Action Plan 2007–2012-Mitigation Program" was approved, and in 2010, the "Andalusian Program for Adaptation to Climate Change" was also approved. These established priority actions to adapt to climate change in energy, the promotion of renewable energy and the promotion of savings and energy efficiency. In addition, in September 2018, The Andalusian Strategy of Circular Bioeconomy was approved, which aimed to contribute to the growth and sustainable development of Andalusia. In order to do so, it sought to promote actions aimed at promoting the production of resources and renewable biological processes, such as olive grove biomass. In this way, this Strategy constituted an economic model, based on the production and use of renewable biomass resources, and their sustainable and efficient transformation into by-products, bioenergy and services for society [76].

Finally, given the relevance of olive cultivation in Andalusia, the agricultural sector has also favored the biomass of olive groves. Proof of this is The Strategic Plan of the Andalusian Agroindustry 2020, which includes nine strategic lines of action in the field of climate change. In this way, it is intended to promote operations aimed at increasing productivity and obtaining better results by consuming less resources, such as water, energy, raw materials, etc., which would be replaced by others with less contaminating impacts (waste, discharges, emissions) [77].

### 3.3.2. Legislation Regulating Olive Biomass in Andalusia

The main regulations that affect olive grove biomass in Andalusia can be classified according to the sector of activity that they regulate (see Table 7). In the legislative field, it should be noted that, in Andalusia, there are no express standards that regulate olive grove biomass. This is regulated, regarding energy matters, by Law 2/2007, which aimed to establish the primacy of the use of renewable energy over other energy sources [78]. This Law includes a set of promotion and exploitation measures for each of the renewable energies. Regarding measures to promote and exploit biomass, Article 17 of this Law establishes the commitment of the Government of Andalusia to regulate the use of energy biomass, develop promotional measures and the valorization process of biomass resources, and to carry out programs to promote energy crops.

**Table 7.** Main regulations of the energy and agricultural sector of Andalusia.

| Activity Sector | Year | Current Regulations |
|---|---|---|
| Energy Sector | 2007 | Law 2/2007,<br>To promote renewable energies, energy savings and efficiency. |
| | 2011 | Decree 169/2011,<br>Regulation of the Promotion of Renewable Energies, Energy Saving and Efficiency approved in Andalusia. |
| | 2018 | Law 8/2018<br>Measures against climate change and for the transition to a new energy model in Andalusia. |
| Agricultural and Forest Sector | 1985 | Law 2/1985,<br>On Civil Protection. |
| | 1999 | Law 5/1999,<br>*Prevention and fighting forest fires.* |
| | 2001 | Law 5/2011,<br>On the olive groves of Andalusia. |
| | 2015 | Decree 103/2015,<br>The Olive Grove Master Plan is approved. |

Source: Own elaboration from BOJA (Official Gazette of the Junta de Andalusia).

Subsequently, Decree 169/2011 developed Law 2/2007. From that moment, the Andalusian Energy Certificate was regulated in a document certifying compliance with energy requirements for buildings and industries. Likewise, a series of obligations were established, such as: the adoption of short-term profitable efficiency and energy saving measures; reliance on renewable energies for the demand of thermal energy; and the establishment of an Energy Management System for large consumers [79].

Given the relevance of the Andalusian olive groves, the autonomous government approved a specific Law for them. Although it went beyond the energy field, it was very favorable for its development. Thus, in 2011, Law 5/2011, relating to the olive groves of Andalusia, came into force. The purpose of this Law was to establish a regulatory framework for the maintenance and improvement of olive growing in Andalusia, the sustainable development of their territories and the promotion of quality and promotion of their products. For this purpose, it was proposed to strengthen the biomass sector, which was also regulated in the said Law [80]. Likewise, following the specifications contained in this Law, in 2015, the Decree 103/2015 (Olive grove Master Plan) was approved. This plan established the specific strategies for olive grove biomass [81]:

- Promote actions aimed at achieving improvements in energy savings and efficiency in olive grove industries.
- Reduce the volume of emissions and effluents in the olive industry.
- Incorporate new technologies for the purification of waste from industries derived from olive groves.
- Value the by-products obtained from olive grove industries and encourage measures for their reuse and/or commercialization.
- Increase the use of renewable energies in the olive grove industries, as well as the energy self-sufficiency of the industries themselves.

As a result, the Olive Grove Master Plan was an opportunity to boost biomass energy in the olive groves, contributing to the 2020 Strategy and the 2014–2020 Rural Development Program.

Law 5/1999, of June 29, on prevention and fighting forest fires and Law 2/1985, of January 21, on Civil Protection, caused olive waste pruning to be destined for production of biomass energy. Law 8/2018 also established a series of measures aimed at reducing greenhouse gas emissions into the atmosphere. For this, the transition towards an energy

model, based on renewable energies, was proposed. This Law sought to promote citizen participation and encourage education, research, development and innovation [82].

Furthermore, at the local level, the different towns and cities of Andalusia joined the Covenant of Mayors. This has promoted the use of renewable energies and, especially, use of olive grove biomass in Andalusia, where 548 participating municipalities (representing more than 40% of the Spanish municipalities taking part) have committed to reducing their $CO_2$ emissions at the municipal level. The intention was to comply with the objectives set by Europe [83]. The signatories of the Covenant of Mayors Pact committed themselves to present an Action Plan for Climate and Sustainable Energy (PACES). This plan detailed the key actions that were intended to be undertaken. Furthermore, within the group of renewable energies, the importance of biomass in the region was highlighted given its huge potential [84]. In short, the olive grove is seen as an important energy alternative to undertake the various projects that are established in renewable energy.

### 3.3.3. Main Promotional Measures for Olive Biomass in Andalusia

The instruments used by the government of Andalusia to promote the use of renewable energies, and, in particular, olive grove biomass, are subsidies for investments in equipment and facilities that use this energy. Since 2003, the Andalusian Energy Agency has been responsible for managing grants. In the case of biomass, it should be noted that in 2006 the Andalusian Society for the Valorization of Biomass was created. Its aims were to develop technologies that would promote the use of energy, as an element of wealth creation in the agricultural and forestry sector. This mixed public–private partnership was the first organization of its kind to be launched in Europe [85].

The most outstanding measures in Andalusia for the promotion of renewable energies, and in particular of olive grove biomass, are shown in Table 8. In it, the most recent measures are listed, although there had previously been other forms of incentives for the promotion of renewable energy and energy efficiency in different economic sectors and households.

**Table 8.** Most recent economic incentives.

| Measures | Approval | Denomination | Description |
|---|---|---|---|
| Promotional measures 2009–2016 (not valid) | Order of 4 February, 2009 | Incentive Program for the Sustainable Energy Development in Andalusia 2009 "Andalusia A +" | Commitment to the promotion of energy saving and use of renewable energies in all areas of Andalusian society. |
| | Decree-Law 1/2014 of 18 March | Program to Promote Sustainable Construction in Andalusia | Integrating several forms of action with incentives aimed at promoting energy saving, improvement of energy efficiency and the use of renewable energy in buildings located in Andalusia. All this was carried out via rehabilitation works, reforms, adaptation to use and efficient installations. Attempted to subsidize operations for obtaining and treating biomass. |
| Promotional measures 2017–2020 (To date) | Order of 23 December, 2016 | Program of Incentives for the Sustainable Energy Development of Andalusia 2020. "Andalusia is more" | The objective of this Program is to continue fostering energy efficiency and the application of renewable resources in the field of building and processes. The intention is to advance energy assessment and management, in sustainable mobility and in implementation of smart networks in the energy field, in line with the Energy Strategy of Andalusia 2020. This continues to subsidize investments aimed at favoring the obtaining and treatment of biomass. |

Source: Own elaboration from Andalusian Energy Agency [86–88].

During the period of validity of the first promotional measures, 2009–2016, 22,949 biomass actions were implemented. These received incentives worth EUR 61 million and, of these, 20,039 corresponded to the Order of Incentives for the Sustainable Energy Development of Andalusia. This aid amounted to EUR 46 million and involved an investment of EUR 118 million. Most of the projects (19,979) were for thermal biomass, followed by logistics and biomass treatment. In addition, since the Sustainable Construction Program in Andalusia came into force in April 2014, and until 31 May, 2016, 2910 thermal biomass actions were supported (stoves, wood fireplaces and boilers) with an incentive of more than EUR 15 million. This led to an associated investment of EUR 23 million [89].

As a result of these actions, there has been a remarkable growth in thermal biomass installations in residential and service sectors. Thus, according to the data provided by the Andalusian Energy Agency, in 2015, biomass consumption in the residential sector was, for the first time, higher than in the industrial sector, with 39.62% compared to 25.84%, represented by the latter [90]. Olive grove by-products played a very important role in this growth.

Subsequently, in order to continue promoting the use of renewable energies in the region, a second program of measures was launched (2017–2020), with the approval of the Order of 23 December, 2016. Thus, through its programs, which were focused on three main points (construction, SMEs and Smart Grids), the intention was to continue promoting the use of renewables, including olive grove biomass. In this way, the different lines of action focused on biomass logistic improvement systems, and energy use in economic sectors and in households.

### 3.3.4. The Olive Grove in the Evolution of Biomass in Andalusia

The consumption of renewable energies in Andalusia is equivalent to 19.48% of primary energy consumption in 2020 (5.2 percentage points above Spain). This contributed to a 43% reduction in carbon dioxide emissions in recent years [75]. However, there is a long way to go to promote the development of renewable energy in the region with a total installed power of 8103.4 MW, compared to 9459.2 MW of fossils. Most of the renewable electric power installed in Andalusia comes from wind energy (3472.0 MW), followed by biomass (274.0 MW), solar thermal (997.4 MW), solar photovoltaic (2672.1 MW) and hydroelectric (650.0 MW). The biomass sector in Andalusia had 2066.2 MW of installed thermal power, which reached 1792.2 MW, as well as electricity, which stood at 274.0 MW [18,75]. This represented 38% of the total installed power for electrical uses, and 21% for thermal uses in Spain.

Figure 5 shows the evolution of installed biomass power for electrical uses in Andalusia, during the period 2000–2020. It can be seen that, since 2000, biomass for electrical uses had not stopped increasing. Since 2005, this had increased by 54.84%. From this moment, there was a greater growth of this energy source. In this sense, it is worth mentioning that the region managed to meet the targets set by PASENER 2007–2013 for biomass for electrical uses, which was set at 256 MW of total installed capacity [73]. Thus, in 2013, biomass for electrical uses stood at 257.5 MW of total installed power, as seen in Figure 5. However, there was a stagnation in the development of this sector from 2012, which was due to national regulatory changes to correct the tariff deficit. In 2019, it was in fifth place with 4.20% of the total renewables installed, generating 1569.2 GWh per year.

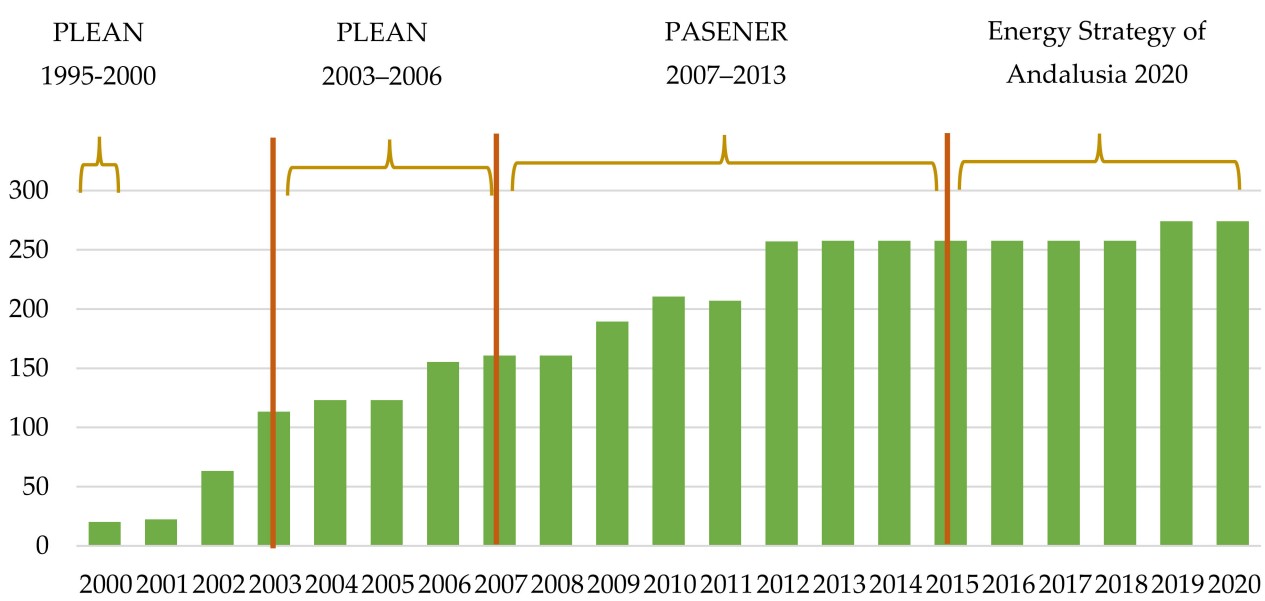

**Figure 5.** Evolution of the electric power of biomass installed in Andalusia during the period 2000–2020 (units: in MW). Source: Own elaboration from Andalusian Energy Agency.

Figure 6 shows the evolution of the installed capacity of biomass for thermal uses during the period 2005–2020. A pronounced increase can be observed starting in 2011, coinciding with the validity of the incentive programs launched during the 2009–2015 period. Thus, since 2010, the installed capacity increased 77.54%. This favored the fulfilment of the objectives set by PASENER 2007–2013 for biomass for thermal uses, which was set at 506.67 MW of total installed power [73]. Thus, in 2013, biomass for thermal uses stood at 1509.2 MW, as can be seen in Figure 6. In 2020 the total power for biomass for thermal uses stood at 1792.2 MW.

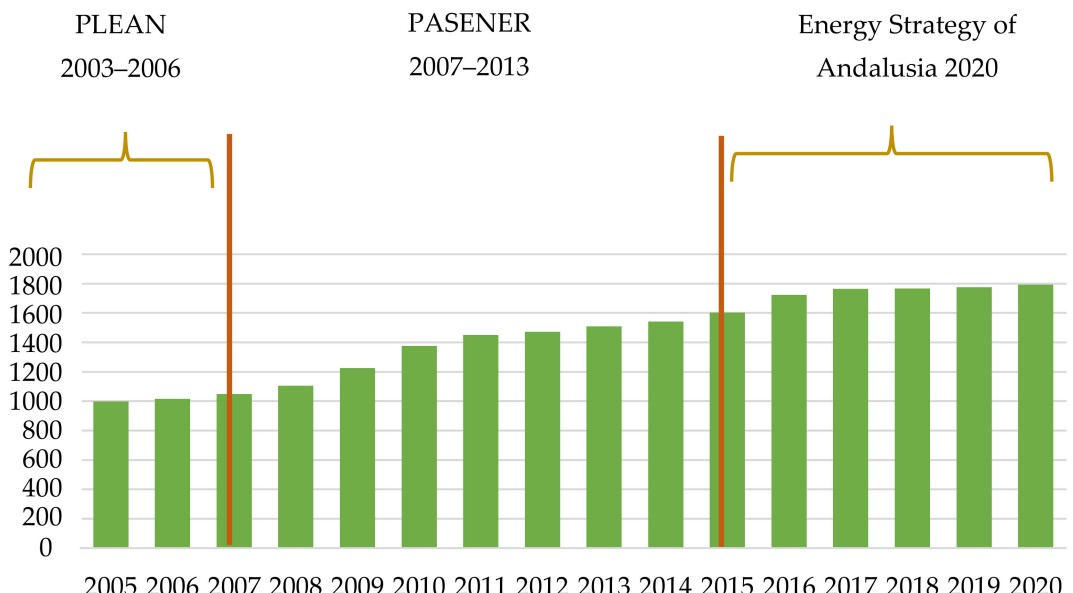

**Figure 6.** Evolution of the thermal power of biomass installed in Andalusia during the period 2005–2020 (units: in MW). Source: Own elaboration from Andalusian Energy Agency.

Increases in the use of biomass had been possible due to the abundant residues generated by olive cultivation and its associated industries in the region. Of the 17 electricity generation facilities in Andalusia, 13 of them used by-products from the olive grove

(135.33 MW of installed capacity). In addition, olive by-products were also widely used for thermal uses [18].

In short, despite the fact that biomass held a privileged place within the renewable energy ranking, there was still considerable waste of resources in this sector. The biomass potential oscillated around 20 million tons per year, with agricultural waste (4.6 million tons) representing the largest volume, followed by industrial waste (5.07 million tons). In both cases, olive grove biomass had a greater participation, with 6.43 million tons, which represented 31.89% of the total biomass potential in the region. Likewise, the potential of olive grove biomass surpassed that of other residues with 1526 ktoe (38.58%). This justified most power generation plants using the main by-products of olive groves as a means of producing renewable energy, as well as the importance of this crop in the territory [18].

## 4. Discussion

The results of this paper show that the implementation of energy plans, laws and promotional measures, to promote the use of olive grove biomass for thermal and electrical uses, have not been sufficient. In Europe, fossil fuels continue to be the dominant energy source, with a consumption of 72.6%. Despite the healthy position that biomass occupies in the renewable energy mix (44.67% of production), it still continues to present a high rate of wasted residues, as is the case of the Mediterranean olive grove. This causes a significant loss of opportunities, when it comes to meeting the objectives set by the different governments, to combat climate change.

In the same way, the regulations, at the national level, analyzed in this paper, reflect the small influence it has had on the development of olive grove biomass as an energy source, for thermal and electrical uses. In this case, the level of biomass development presents a less favorable situation than at the European level. This energy source only generates 5% of the total generated by renewable energies, despite the high potential of olive grove biomass. Currently, in Spain, the most developed renewable energies are wind (with an installed power of 22,990 MW) and hydro (with an installed power of 18,963 MW). Biomass, for its part, only represents 1.34% of installed power of renewable energies, for electrical uses (with 677 MW), and 16.42% for thermal uses (with 8297 MW). This is largely due to the low remuneration received, in line with other renewable energies that are in a more privileged situation, such as solar photovoltaic, solar thermoelectric and wind energy. Likewise, the olive grove sector faces other obstacles, such as the cost of labor and transport, in the use of residues.

Despite the large area occupied by olive groves in Andalusia (60% at the national level and 30% at the European level), the development of this energy source follows the same dynamics as at the national level, with only 274.0 MW of installed power for electrical uses, and 1792.2 MW of installed power for thermal uses. However, it should be noted that, over the years, this use of this energy source has been increasing for thermal and electrical energy, experiencing its greatest development since 2005. This trend implied the fulfillment of the objectives set by the PASENER 2007–2013 for thermal and electrical applications. However, despite this, the wastage of this organic matter is still high. Since 2012, with the regulations applied at the national level, when the system of feed-in tariffs for renewable energies began to be withdrawn, the development of biomass and olive grove biomass has been stagnant.

In this sense, more favorable regulations and greater incentive for olive grove biomass would allow promoting the use of this energy source in the places where this crop is grown. Thus, countries and regions with a large number of olive groves could take advantage of the opportunities that this crop offers them in terms of energy, as reflected in numerous investigations. In this way, focusing on the paper of Marquina et al. (2021) [91], it is possible to highlight the high wastage of olive grove biomass in the region. Specifically, this represents 69.23% of waste, causing lost opportunities that translate into less development of renewable energies and, especially, of olive grove biomass. In addition, Marquina et al. (2021) [91] establish that a greater use of olive grove biomass would bring about the start-up

of 55 and 1762 electric and thermal plants, respectively. This would cause a further increase in investment and, consequently, an opportunity for the creation of new jobs, which would represent a positive aspect for the regional economy affected by the economic crisis caused by COVID-19. Likewise, the potential for olive grove biomass has also been the object of studies in other research works, such as that of García-Maraver et al. (2011) [17] for Andalusia, and that of Algieri et al. (2019) [92], Spinelli et al. (2011) [93], Alatzas et al. (2019) [94] and Dounavis (2019) [95] for Italy and Greece.

In this sense, a better use of the olive grove biomass would result in a greater reduction in greenhouse gases, thus promoting a more sustainable economy, based on the bioeconomy and circular economy. This would tend to better fulfill the objectives set in the different energy plans, whilst promulgating a Green Recovery in the face of the economic crisis as a result of COVID-19. Thus, given that current economic situation, a greater use of waste from olive groves and the olive agro-industrial sector would bring with it greater economic development, especially in rural areas where this crop is grown. However, it is convenient to mention that olive grove biomass, in addition to presenting opportunities (being renewable; reducing dependence and consumption of fossil fuels; contributing to the reduction of residues; or having a neutral balance in $CO_2$ emissions), also presents a series of negative repercussions, such as: emitting $CO_2$ into the atmosphere; high labor and transportation costs; its supply depends on seasonality; and high costs in the construction and operation of plants [96].

Furthermore, it is also worth highlighting the COVID-19 lockdown on energy demand and consumption in Europe. Thus, according to Jiang et al. (2021), compared to the mean value from 2015 to 2019, the total mean electricity generation from 16 European countries in April 2020 dropped by 9% (25 GW), where fossil energy generation decreased by 28% (24 GW), nuclear energy decreased by 14% (11 GW), whilst renewables increased by 15% (15 GW) [97]. In this sense, it should be noted that despite the COVID-19 crisis, the objectives of complying with an economy that respect the environment remain a priority.

In short, based on the results obtained, it is worth highlighting the limited benefit that the regulations in force until 2012 have shown and the high obstacle presented by the regulations approved as of that year despite the high potential presented by Andalusia. For this reason, it would be convenient to change this regulatory aspect towards a more favorable aspect with clean energies and, especially, with olive grove biomass. Thus, the ultimate goal would be to promote a green recovery to incentivize renewable energies while tending to stimulate the economy in general. The aim of this work is to show how the main regulations have influenced the development of olive grove biomass. Thus, the benefits that this research presents is to give knowledge of how the main regulation of renewable energies has influenced and influences their development and to highlight the losses of opportunities that this reports.

## 5. Conclusions

At present, the current energy regulations at the national level represent a strong barrier for the advancement and development of the consumption of biomass, in particular from the olive groves and associated industries. The waste of this organic matter, at the regional level, represents 69.23%. If all the waste were used, it would translate into more generation of electrical and thermal energy.

Over the years, the approval and implementation of different laws, plans and incentives, both nationally and regionally, have not been sufficient for the development of olive grove biomass. Thus, in Andalusia, although the olive grove biomass had an increase in electrical and thermal uses from 2005, until it reached the PASENER 2007–2013 objectives, this situation was hampered by the approval of different regulations, approved in 2012. In this sense, Royal Decree-Law 1/2012 and Royal Decree-Law 2/2013 represented a setback for the development of olive grove biomass. From that moment on, the feed-in tariff system, which was an advantage for these energy sources, was abolished.

Similarly, this sector has other obstacles (such as transport and labor, in the case of olive pruning) which have not been corrected. Incentives are still scarce, and growers do not find it profitable to collect prunings for later use. This is due to the low rewards, which do not even cover collection costs. In addition to all this, there is the lack of information on the possibilities of using this organic matter.

Regarding the promotion of biomass in the generation of electricity, it has been observed how the regulations have not been able to correct the difference between this energy source, and the other renewable energies. The need to acquire fuel (olive grove biomass) requires sufficient remuneration to make the heavy investment profitable, and makes it less competitive than wind or photovoltaic. As a result, biomass for electrical uses has only reached 677 MW of installed power in Spain. This is a small development considering the expectations established in each of the energy plans for this energy source.

To correct this situation, it would be necessary to carry out specific electricity auctions of biomass power generation plants, nationally, providing adequate remuneration to ensure the profitability of the projects. Regarding Andalusia, to promote olive grove biomass, specific measures for this sector need to be established among growers, in order to promote environmental practices that make it possible to obtain biomass. Among these measures would be training and advising growers, subsidizing investment in machinery, and promoting agricultural service companies specialized in obtaining biomass from olive groves.

However, in terms of biomass for thermal uses, it would appear that regulations, such as the inclusion of renewables in buildings, or existing national and regional subsidy programs, would lead to greater development of this technology. This makes it one of the most widely used technologies in the residential sector, although the implementation of regional measures would further increase its use.

In future national and regional energy plans, it would be considered appropriate to introduce the adequacy of the use of different types of biomass for thermal or electric uses, depending on the value chain of each of these uses, and to promote measures that contribute to compliance with the established objectives.

In short, although Europe has tried to favor the biomass sector, more especially since 2005, Spain has not had the necessary means to overcome the barriers that this sector presents. Thus, Spain, and especially Andalusia, are wasting an opportunity from the energy and economic point of view. In this sense, this paper opens the debate and joins the groups that are committed to a green recovery in the face of the economic crisis caused by COVID-19.

**Author Contributions:** Data curation: J.M. and M.J.C.; Formal analysis: J.M.; Methodology: J.M., M.J.C. and M.d.P.P.-R.; Writing—Original draft preparation: J.M.; Resources M.J.C. and M.d.P.P.-R.; Supervision: M.J.C. and M.d.P.P.-R.; Conceptualization: M.d.P.P.-R.; Writing—Reviewing and Editing: M.d.P.P.-R.; Funding acquisition: M.d.P.P.-R. All authors have read and agreed to the published version of the manuscript.

**Funding:** This research was funded by the Chair on Energy and Environmental Economics sponsored by "Red Eléctrica de España" at the University of Seville (Ref; 68/83-1394-0103) and by Fondo Europeo de Desarrollo Regional (FEDER) and Consejería de Economía, Conocimiento, Empresas y Universidad of Junta de Andalucía, on FEDER 2014–2020 operative programme: reference-(US-1260925) US/JUNTA/FEDER, UE.

**Institutional Review Board Statement:** Not applicable.

**Informed Consent Statement:** Not applicable.

**Data Availability Statement:** The data presented in this study are available upon request.

**Conflicts of Interest:** The authors declare no conflict of interest.

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
