# Peer review of "Measures to Promote Olive Grove Biomass in Spain and Andalusia: An Opportunity for Economic Recovery against COVID-19"

_sustainability, doi:10.3390/su132011318_

Round 1

Reviewer 1 Report

After having assessed the suitability for publication of the Manuscript ID: sustainability-1392043, having the title "Measures to promote olive grove biomass in Spain and Andalusia: An opportunity for economic recovery against COVID-19", submitted to the "Energy Sustainability" section of the MDPI Journal Sustainability, I have distinguished several elements that from my point of view should be made less confused and more comprehensible by the authors in view of improving the quality of the manuscript. Therefore, I have devised and wrote a series of comments to the authors of the manuscript under review.

In this paper, the authors analyze the evolution and current situation of the regulations on the production of biomass energy that affects olive grove residues in Andalusia (Spain), the analysis being carried out from European, national and regional perspectives.

 The Manuscript ID: sustainability-1392043 is interesting and generally well-structured. However, the article under review will be improved if the authors address the following aspects in the text of the manuscript and reflect them clearly point-by-point within the coverletter:

  1. The main weak point of the manuscript. The authors have submitted their manuscript by specifying the type of their paper as "Article". However, at Lines 119-120, the authors stated: "In order to develop this paper, a systematic review of the main regulations governing olive grove biomass in Europe, Spain and Andalusia, has been carried out." Therefore, by taking into consideration this statement and also the content of the manuscript, it seems that the manuscript is more suitable and has the potential of becoming a "review" article type rather than an "article" type that implies bringing novel contributions to the current state of knowledge. The authors should address this inconsistency by deciding what is the type of their paper. Even if the authors decide that their paper is a review one, or they decide that their paper is a research one (based on a systematic review), starting from the above-mentioned "systematic review", the authors should emphasize more in the manuscript (eventually in an Appendix), along with the elements already presented, the literature search method and criteria, the main search engines and used keywords, the approach that they have used in developing their study and what are the characteristics that allow them to consider their study as being "a systematic review" when compared to other review studies from the scientific literature and why their paper could not be considered as another type of review (for example Rapid Review, Scientific Literature (Narrative) Review, Scoping Review, Mixed Methods Review, Network Meta-Analysis, Overview of Reviews, Meta-Syntheses, Integrative Review, Diagnostic Test Accuracy Review, Living Systematic Review). Taking into consideration that in the manuscript it is specified the type "Article", I had to devise my review comments in rapport with the type "Article" that implies that the authors must bring original contributions to the current state of knowledge. Consequently, the authors must assume clearly their original contribution by specifying this fact and by highlighting the fact that starting from a certain point there are presented the original and novel aspects of their research. The authors must assume clearer their own methods, own results, own conclusions, the novelty of their study and own original contribution to the current state of knowledge.
  2. Lines 9-22, the "Abstract" of the paper. The manuscript will benefit if, along with the elements already presented, the authors describe in the abstract the background (in which the authors should place the issue that the manuscript addresses in a broad context and highlight the purpose of the study), in order to provide a structured abstract containing: the background, the methods, the main findings, the conclusions, as in the actual form of the manuscript, the abstract offers information related only to some of these aspects and even so, their delimitation is unclear. In the abstract the authors must also declare and briefly justify the novelty of their work.
  3. The "Introduction" section – the gap in the current state of knowledge. After having presented the literature survey, the authors should pinpoint an exact deficiency, an unsolved problem that still exists in the current body of knowledge that their study addresses. This aspect will improve the manuscript under review on multiple plans, as the identified deficiency, the identified unsolved problem will offer great opportunities to highlight and prove, when discussing their results, the contribution, the advancement that the conducted research has brought to the existing state of knowledge. Afterwards, it will benefit to state the novel aspects of the conducted study.
  4. The "Introduction" section – the main original contributions of the paper. It will benefit the paper if in the final part of the "Introduction" section, the authors present the main original contributions of their paper, eventually synthetized within a bulleted list.
  5. The "Methodology" section - presenting the devised approach. In order to help the readers better understand the methodology of the conducted research, in addition to the figures already presented, the authors should devise a flowchart when presenting their approach, a flowchart that depicts the steps that the authors have processed in developing their research and most important of all, the final target. This flowchart will facilitate the understanding and the reproducibility of the proposed approach and at the same time will make the article more interesting and help promote it to the readers.
  6. The "Methodology" section - details regarding the devised systematic review. I consider that the methodology lacks an aspect of paramount importance that is mandatory when devising a systematic review, namely, to present the used databases, the full electronic search strategy (search queries) for querying the databases, the date and time when the search queries that gathered the initial pool of scientific works were run, including any limits used, such that it could assure the reproducibility of the study. In the current form of the manuscript, the databases and the electronic search performed by the authors on them cannot be reproduced as the search queries comprising the search keyword are missing. In addition, in the current form of the manuscript, the methodology does not provide the rationale for the eligibility criteria, a complete rationale for the study design.
  7. The "Results" section - details regarding the locations. At Lines 455-461, the authors state: "The high potential of biomass in Spain is due, in part, to the cultivation of the olive and the waste generated by the agro-food olive oil industries. The territory has 2.65 million hectares occupied by this crop. These are mainly concentrated in Andalusia (60.38%), although they are also present in Castile-La Mancha (15.84%) and Extremadura (10.49%) [13]. Thus, Spain has large areas of olive cultivation where around 282.70 million olive groves are concentrated. This generates large amounts of biomass per year, as a result of the different olive grove pruning processes (mainly, thin branches and firewood) [62].)." It will benefit the manuscript if, along with these details, the authors provide a detailed image, containing both the layout and the contours of the terrain, highlighting the targeted locations eventually using as source the Google Maps web mapping service or any other similar approach.
  8. The "Results" section - details regarding the datasets. In what concerns the datasets, the authors state at Lines 436-663: "Figure 2 shows the evolution of total installed renewable power, as well as the power in biomass, for both thermal and electrical uses, from the period 1998-2004 to the period 2013-2016. The periods mark the time that each of the main Royal Decrees, related to the production of renewable energies, had been in force (RD 2818/1998, RD 436/2004, RD 661/2007 and RD 413/2014). It can be observed that the installed biomass power, like the other renewable energies, had greater growth during the validity of PER 2005-2010, with increases of more than 70% with respect to the period 2004-2007, when the PFER 2000-442 2010 was in force." In this context, the authors should comment in the paper whether the data collected in the period 1998-2004 are still relevant today, in 2021, in what concerns the same analyzed parameters. The authors should provide explanations whether their study is consistent, whether the changes that may occur within the older dataset from the above-mentioned years and the current year risk altering the final result. The authors should take into account the fact that a part of the information might change in time, or information can be outdated and therefore the whole study risks to become inconsistent and irrelevant. In addition to these, the authors must provide more details regarding the way in which they intend to solve the problems related to missing data or abnormal values if they are to occur.
  9. The "Discussion" section, Lines 725-777. Even if the authors have cited a paper (namely [85]) when discussing their obtained results, this citation is used exclusively in order to sustain and justify the statements from the manuscript under review, but not in the purpose of devising a comparison between the authors' obtained results / devised approach and other existing ones from the literature. Therefore, this section of the manuscript does not reflect in a clear manner the way in which the obtained approach can be perceived in perspective of previous studies that have tackled similar problems. This comparison is mandatory in order to highlight the clear contribution to the current state of knowledge that the authors have brought. The authors should underline both the advantages and disadvantages of their proposed approach when compared with other valuable studies from the current state of art. When discussing their obtained results, the authors should emphasize not only the novel aspects and strong points of their developed method, but also should point out objectively the existing limitations of their method, possible circumstances that will hinder their method’s effectiveness and state clear and accurate directions they will pursue in their future research activities in order to extend the current research and overcome these limitations.
  10. Insights. The paper will benefit if the authors make a step further, beyond their approach and provide an insight at the end of the "Discussion" section regarding what they consider to be, based on the obtained results, the most important, appropriate and concrete actions that the decisional factors and all the involved parties should take in order to benefit from the results of the research conducted within the manuscript as to attain the ultimate goal of sustainability and also the most important benefits of the research conducted within the manuscript, taking also into account its practical applicability.
  11. The impact of the COVID-19 lockdown. As the Manuscript ID: sustainability-1392043 approaches issues regarding the economic recovery against COVID-19 based on biomass energy, I would like the authors to discuss in the paper if and how their developed approach can be adjusted as to take into consideration the impact of the COVID-19 lockdown on energy demand and consumption. Regarding this impact, in the paper "Jiang, Peng et al., Impacts of COVID-19 on energy demand and consumption: Challenges, lessons and emerging opportunities, Applied energy, vol. 285 (2021): 116441. doi:10.1016/j.apenergy.2021.116441", one can find the following statements: "Compared to the mean value from 2015 to 2019, the total mean electricity generation from 16 European countries in April 2020 dropped by 9% (25 GW), where fossil energy generation decreased by 28% (24 GW), nuclear energy decreased by 14% (11 GW), whilst renewables increased by 15% (15 GW)." The authors should comment in their paper the above-mentioned impact, eventually referring to the above-mentioned article (or to any other scientific papers they consider, related to this subject).
  12. The advantages and disadvantages of using biomass energy. In the reviewed manuscript, the authors approach issues regarding the "measures to promote olive grove biomass". However, even if this type of energy has unquestionable advantages (such as: being renewable; reducing the dependence and consumption of fossil fuels; being carbon neutral; contributing to waste reduction), it also has a series of negative impacts (like: the carbon dioxide emissions; the high costs of the construction and operation of biomass energy plants, of the extraction and transportation of biomass materials; the seasonality of biomass supply). It will benefit the paper if the authors comment in their research these pros and cons.

Minor remarks.

  • The title of the "Conclusions" section. I have noticed that the authors have chosen to entitle the section 5 as "Conclusion" but obviously, this section depicts more than one conclusion. Therefore, the title "Conclusions" is more suitable (just like the official Sustainability MDPI Journal's Template recommends).
  • The Figures within the manuscript. The authors must specify the measuring units (when applicable) along with the axes' titles within the figures.
  • The acronyms within the paper. At Lines 15-16, the authors state: "…reaching the objectives set by the PASENER 2007-2013…" Even if it is widely known in the scientific community, the PASENER acronym, as well as any other acronyms, should be explained the first time when they appear in the manuscript, for example, in the above-mentioned case, under the form: "…reaching the objectives set by the Andalusian Energy Sustainability Plan (Plan Andaluz de Sostenibilidad Energética - PASENER) 2007-2013…".

Author Response

We thank the reviewer´s comments. Please see the attached document where the changes are pointed out. 

Reviewer 2 Report

  1. The topic is very interesting and the autor's idea has the scientific sound, it summarizes the years of work on legislation related to renewable energy in Europe and especially in Andalusia. Considering that it is a hot problem and a lot of papers are created each month, it is unacceptable that the data and references are not up to date. For Authors currently means year 2018 (page 9, line 360: "currently..." and referrence to literature number 46 which is accessed September 2018 and the latest data available then was year 2016). Many literature items are accessed two-three years ago (ex. no 6, 10-12, 32, 34, 36, 37, 40-42, 62-63, 70-71,77, 79), and Eurostat statistics are accessed January 2020, while the current date is September 2021, it is more than 1,5 year of delay. It may no influence the historical data used in the paper or changes in law described, but the paper about renewable energy should consider that in such area we have dynamic change year to year. Since 2016 or 2017 situation can be different, maybe not so dramatically, but there is still a gap between 2017 and 2021. The paper should be rewrite with the latest available data which is at least 2019.
  2.  In 2019 European Parliament launched the Green Deal (COM(2019)0640), which should be included to the paper and also in 2020 it is issued annex on guidelines for trans-European energy infrastructure COM(2020)0824 which also could be taken into consideration.
  3. As I understand, all pages with the calculated values refer to the year 2016 or 2017? For example page 20 line 667: "The consumption of renewable energies in Andalusia is equivalent to 20.6% of primary energy consumption (5.5 percentage points above Spain)" and the references to this is numer 79 accessed September 2019. So for which year is that percentage? Maybe today is more or less the same situation but this exact number 20.6% should be given for specific period.

Author Response

We thank the reviewer´s comments. Please see the attached document where the paper changes are remarked 

Reviewer 3 Report

Thank you for your revisions. While I believe the review of regulations in relation to a particular sector is important, I still do not believe the authors convince me of the originality of this work or the contribution to the wider literature. I'm afraid the paper reads too much like a report.

Author Response

We thanks the reviewer effort and invite him/her to read the new version of the paper

Round 2

Reviewer 1 Report

After having assessed the suitability for publication of the revised version of the Manuscript ID: sustainability-1392043, having the title "Measures to promote olive grove biomass in Spain and Andalusia: An opportunity for economic recovery against COVID-19", I can conclude that the authors have addressed the most important signaled issues, therefore improving the manuscript in contrast to the prior submission.

Reviewer 2 Report

All reviewers remarks has been correctly adressed.

Reviewer 3 Report

I am content that effort has been made to address my comments. Thanks to the authors for doing this.

This manuscript is a resubmission of an earlier submission. The following is a list of the peer review reports and author responses from that submission.

Round 1

Reviewer 1 Report

Overall I wasn't sold on the merits of this paper in 3 different dimensions.

  • I did not feel it demonstrated to me what literature it was building upon and what contribution it was making
  • The research questions were not clearly elucidated, nor was the theoretical perspective and hypotheses
  • As a consquence the analysis was disjointed from the introduction
  • As a result, the paper reads like a normative policy proposal rather than a critical scientific journal article

To some extent this read like a proposal for more use of olives for biomass purposes, but without a critical assessments of the merits of using land for food versus other uses.

I felt the link with COVID was a bit tenuous. Rural economic development is important in any time.

Areas that could improve the paper

  1. Define more clearly what the contribution is. What does the literature have to say about policies to promote biomass creation. Are there specific gaps in the literature. This was not clear to me in this paper.
  2. Are there alternative uses of the land, sources of food or sources of energy. Given the location of the olive groves, are there not more opportunities for zero emissions energy?
  3. While there was a single reference to the circular bio-economy., there was very little discussion and only passing references to waste by-products. Given the amount of waste generate, why are they not being used more? If it is proposed to increase the hectareage, what are the deciding factors versus existing land uses.
  4. It would have been useful to develop a theoretical framework to consider the purpose of the paper. Is it discuss the opportunities of energy use or to discuss the political framework. There are ample theoretical perspectives of either perspective. At present there are a list of things that should be done that are stated without critique. A useful approach the authors might find, might be to look at the innovation system required to facilitate developments and to frame the analysis of recommendations in relation to the functions and behaviours of different Innovation System Actors
  5. I would recommend that a number of key take-home points are agreed and that a clear narrative would build from context through theory to method to analysis and discussions. The paper does not build in that way and a result it it difficult to assess the justification of the recommendations in the conclusions

Reviewer 2 Report

In this study, the authors  the analyses the evolution and current situation of the regulations on the production of biomass energy that affects olive grove residues in Andalusia (Spain). The analysis is carried out from a triple perspective; European, national and regional. For this, plans, programs, laws and other approved regulations are analyzed to highlight their main characteristics. Likewise, we analyze in what way these measures have influenced the development of biomass from olive groves for electrical and thermal uses in Andalusia.
The manuscript represen the opinion of Jesus Marquina  et al., and confirme that  the results of the analysis show that attempts to encourage biomass for such uses have not been sufficient. In addition, the regulations approved in recent years by the national government have been a great barrier to its development. Furthermore, the olive grove biomass sector has other obstacles that have not favored its use for energy, such as the costs of residues collection and the few incentives for this sector. This fact has caused a lower energy development in this sector compared to other renewable sources. Also, it should be pointed out that the reorientation of measures, with the purpose of boosting this energy source, would supply a positive effect for the economy of the region which has been greatly affected by the health crisis caused by COVID-19.
This may be ambitious and interesting, but some important points need to be resolved. Importantly, a study must provide a critical analysis of the data. In other words, you must assess whether specific data published really stand up to scientific scrutiny. In order to achieve the above, you must clearly define your specific aims and objectives. So in your study you must develop a critical appraisal of the state of the art. This is an essential element of any  article. There are important scientific questions (both conceptual and methodological) which need to be addressed with the primary studies. A study must highlight this. The introduction, which is written in clear language, covers a large number of relevant issues. Information are noteworthy, and are correct supported by similar results from the specialty literature Try to rewrite the abstract  and conclusions, I also recommend the nuance of the introduction